# Modelled changes in source contributions of particulate matter during the COVID-19 pandemic in the Yangtze River Delta, China

Jinlong Ma[1], Juanyong Shen[2], Peng Wang[3], Shengqiang Zhu[1], Yu Wang[1], Pengfei Wang[4], Gehui Wang[5,6], Jianmin Chen[1,6*], Hongliang Zhang[1,6*]

[1]Department of Environmental Science and Engineering, Fudan University, Shanghai 200438, China
[2]School of Environmental Science and Engineering, Shanghai Jiao Tong University, Shanghai 200240, China
[3]Department of Civil and Environmental Engineering, Hong Kong Polytechnic University, Hong Kong 99907, China
[4]Department of Civil and Environmental Engineering, Louisiana State University, Baton Rouge, LA 70803, USA
[5]Key Lab of Geographic Information Science of the Ministry of Education, School of Geographic Sciences, East China Normal University, Shanghai 200241, China
[6]Institute of Eco-Chongming (IEC), Shanghai 200062, China

*Correspondence to*: Hongliang Zhang (zhanghl@fudan.edu.cn); Jianmin Chen (jmchen@fudan.edu.cn)

**Abstract.** Within a short time after the outbreak of coronavirus disease 2019 (COVID-19) in Wuhan, Hubei, the Chinese government took the nationwide lockdown to prevent the spread of the pandemic. The quarantine measures have significantly decreased the anthropogenic activities, and thus improving air quality. To study the impacts caused by the lockdown on specific source sectors and regions in the Yangtze River Delta (YRD), the Community Multiscale Air Quality (CMAQ) model was used to investigate the changes in source contributions to fine particulate matter ($PM_{2.5}$) from January 23 to February 28, 2020, based on different emission control cases. Compared to Case 1 (without emission reductions), the total $PM_{2.5}$ mass for Case 2 (with emission reductions) decreased by more than 20% over the entire YRD and the reduction ratios of its components were 15%, 16%, 20%, 43%, 34%, and 35% in primary organic aerosol (POA), elemental carbon (EC), sulfate, nitrate, ammonium, and secondary organic aerosol (SOA), respectively. The source apportionment results showed that $PM_{2.5}$ concentrations from transportation decreased by 40% while from the residential and power sectors decreased by less than 10% due to the lockdown. Although all sources decreased, the relative contribution changed differently. Contributions of the residential sector increased by more than 10% to 35%, while those in the industrial sector decreased by 33%. Considering regional transport, the total $PM_{2.5}$ mass of all regions decreased 20-30% in the YRD with the largest decreased value of 5.0 μg m$^{-3}$ in Henan, Hebei, Beijing and Tianjin (Ha-BTH). In Shanghai, the lower contributions from local emissions and regional transmission (mainly Shandong and Ha-BTH) led to the reduced $PM_{2.5}$. This study suggests adjustments of control measures for various sources and regions.

## 1 Introduction

Fine particulate matter ($PM_{2.5}$, an aerodynamic diameter of fewer than 2.5 μm) has been a great concern in China since 2013 due to its high levels and related health risks (Lelieveld et al., 2015;Huang et al., 2014;He and Christakos, 2018;Shang et al., 2018;Song et al., 2017;Song et al., 2016;Yan et al., 2018;Du and Li, 2016;Liu et al., 2016;Shen et al., 2020a). To improve air quality, China has promulgated stringent emission control plans such as the Air Pollution Prevention and Control Action Plan and $PM_{2.5}$ concentrations have been reduced significantly in different regions (Zheng et al., 2018;Cai et al., 2017;Zhang et al.,

2016;Zheng et al., 2017). In the Yangtze River Delta (YRD), one of the largest economic centers, $PM_{2.5}$ concentrations were reduced by 34.3% from 2013 to 2017 due to significant efforts (China, 2018). However, $PM_{2.5}$ concentrations are still much higher than the recommended annual mean criteria of 10 $\mu g\ m^{-3}$ by the World Health Organization (WHO). The significant reductions in emissions lead to changes in the local and regional transport contributions of key pollutants. Consequently, the air quality strategies need further improvement according to the source apportionment results.

$PM_{2.5}$ is a complex mixture of primary PM components (PPM) and secondary formed components, and its source apportionment is based on quantifying the contributions of different sources to all the components. Statistical methods based on observed $PM_{2.5}$ composition information using source profiles of different emission sources and assuming that composition remains unchanged in the atmosphere can only resolved contributions of different source sectors to PPM, leaving secondary components as a whole (Tao et al., 2014;Gao et al., 2016;Yao et al., 2016;Zhang et al., 2013;Zhu et al., 2018). Source-oriented chemical transport models (CTMs) are capable of investigating the contributions of both source sectors and regional transports to both PPM and secondary components (Wang et al., 2014;Ying et al., 2014;Wang et al., 2013;Yang et al., 2020). For instance, Hu et al. (2015) reported that local emissions accounted for the highest fraction of PPM compared to the regional transport in Shanghai. Zhang et al. (2012) showed that the power sector (~30%) was the predominant contributor to sulfate, a component of secondary inorganic aerosol (SIA), and the remaining contributions were from industrial and residential sectors in Shanghai. Liu et al. (2020) reported that the industry sector was the major secondary organic aerosol (SOA) emissions source, and additionally both regional transport and local emissions were critical to Shanghai. With source contributions changed, the information provided by these studies is not suitable for the further reduction of $PM_{2.5}$ in the YRD. Therefore, updated source apportionment information is needed to support the formulation of further reduction policy.

To prevent the spread of the COVID-19 pandemic, the unprecedented nationwide lockdown has been implemented to limit anthropogenic activities since January 2020. As a result, anthropogenic emissions decreased drastically, especially in the transportation and industry sectors (Wang et al., 2020a). As a natural experiment with high research values, this provides a valuable opportunity to understand pollution changes with extremely strict measures. Studies have reported significant decreases of $PM_{2.5}$ in the YRD based on absolute concentrations (Chen et al., 2020;Li et al., 2020;Chauhan and Singh, 2020;Yuan et al., 2020). However, it is not clear how the contributions of local sources and regional transport changed, and the conclusions reported in the mentioned literature cannot be used to design control strategies. Thus, it is critical to investigate changes in source sectors and regions during the COVID-19 pandemic.

In this study, a source-oriented version of the Community Multiscale Air Quality (CMAQ) model is used to determine the contributions of source sectors and regional transport to $PM_{2.5}$ in the YRD from January 23 to February 28. The impacts of quarantine measures are estimated by comparing the contributions before and after January 23, the start point of the lockdown. The results offer a deep insight into $PM_{2.5}$ source changes and help develop suitable emission control measures.

## 2 Methodology

## 2.1 Model description

The SAPRC-11 photochemical mechanism and AERO6 aerosol module are applied in the CMAQ v5.0.2 to separately quantify source contributions to PPM and SIA (Carter and Heo, 2013;Zhang et al., 2015). The CMAQ model used in this study was modified with additional non-reactive tracers of PPM from various source sectors and regions (Hu et al., 2015). The emission
rates of these tracers only account for 0.001% of total PPM emission rates in each grid cell so that they will not have an impact on the atmospheric process, as shown in equation (1):

$$ATCR_i = 10^{-5} * PPM_i \tag{1}$$

where $ATCR_i$ represents emission rate of the tracer from the $i$th emission source or region with PPM emission rate of $PPM_i$, and $10^{-5}$ is the scaling factor. The concentrations of tracers from a given source or region are then estimated by
multiplying $10^5$ to represent the concentrations of PPM from that source or region. The concentrations of components in PPM are calculated based on the ratio of each component to total PPM from sources or regions. Details were discussed in Hu et al. (2015).

The contributions of source sectors and regions to SIA are quantified by tagging reactive tracers. Precisely, both the components of SIA and their precursors from diverse source types and regions are tracked separately by adding labels on $NO_x$,
$SO_2$, and $NH_3$ through the atmospheric process (Shi et al., 2017). In this study, contributions from different emission sectors including residential, industry, transportation, power, and agriculture, and those from source regions including Jiangsu, Shanghai, Zhejiang, Anhui, Ha-BTH (Henan, Hebei, Beijing, and Tianjin), Shandong, HnHb (Hunan and Hubei) and other provinces are tracked (Fig. S1 and Table S1). The SOA simulation has considerable uncertainties, which were caused by the inadequate knowledge of its precursors, incomprehensive formation mechanisms in the model, and limited observations (Zhao
et al., 2016;Yang et al., 2019;Heald et al., 2005;Carlton et al., 2008). Therefore, the SOA sources are not tracked in this study. More information of SOA source apportionment was discussed in Wang et al. (2018).

## 2.2 Model application

Two nested domains were used to simulate pollution changes during the COVID-19 pandemic from January 5 to February 28, 2020. As shown in Fig. S1, China and its surrounding areas are covered in the outer 36 km domain (197 × 127 grid cells), and
the YRD is covered by the inner 12km domain (97 × 88 grid cells). The first five-day simulation is removed to minimize the effect of initial conditions. The boundary conditions used in the 12 km domain are offered by the 36 km simulations. Meteorology inputs were generated by the Weather Research and Forecasting (WRF) model v3.6.1. The boundary and initial conditions for WRF were from the National Centers for Environmental Prediction (NCEP) Final (FNL) Operational Model Global Tropospheric Analyses dataset (available at http://rda.ucar.edu/datasets/ds083.2/). The anthropogenic emissions in
China, based on the Multi-resolution Emission Inventory for China (MEIC) (http://www.meicmodel.org), include industry, power, agriculture, residential and transportation. The emissions from other countries were obtained from the Emissions Database for Global Atmospdheric Research (EDGAR) v4.3 (http://edgar.jrc.ec.europa.eu/overview.php?v=431). Biogenic

emissions were generated using the Model for Emissions of Gases and Aerosols from Nature (MEGAN) v2.1 (Guenther et al., 2012;Guenther et al., 2006).

Two cases were simulated in this study (Table 1). The base case (Case 1) used the original inventory. In Case 2, the emissions of carbon monoxide (CO), nitric oxide ($NO_x$), sulfur dioxide ($SO_2$), volatile organic compounds (VOC), and PM were decreased during the COVID-19 period since January 23 with provincial specific factors from Huang et al. (2020). The differences between the cases represent the changes in sources and regions.

## 3 Results and discussion

### 3.1 Model performance

#### 3.1.1 WRF evaluation

Since air quality simulations would be influenced by meteorology difference, it is critical to validate WRF performance before simulating source apportionment (Zhang et al., 2015). The model performance of meteorological parameters including temperature at 2 m above the ground surface (T2), wind speed (WSPD), wind direction (WD) and relative humidity (RH) in the COVID-19 period could be found in Table S2. The statistical values of mean prediction (PRE), mean observation (OBS), mean bias (MB), gross error (GE), and root mean square error (RMSE) have been calculated and the calculation formulas were listed in Table S4. T2 predicted by the WRF model were slightly higher than observations in the two periods. The MB values of T2 before and after the lockdown were both 1.6, while the GE value of T2 before the lockdown period was slightly larger than the recommended criterion based on Emery et al. (2001). Except for the MB values of WSPD, both GE (1.3 and 1.6) and RMSE (1.7 and 2.0) met the benchmarks during the two periods. The MB (1.8) and GE (29.2) values of WD were all within the benchmarks after the lockdown, but the GE value of WD before the lockdown was slightly higher than the benchmark. The simulated RH was underestimated with the MB values of -2.4 and -5.6 during the two periods. The hourly comparisons of T2, WSPD and RH were shown in Fig. S9 based on Wang et al. (2020b, under review) also indicated good model performance. Compared to previous studies (Chen et al., 2019;Liu et al., 2020), the meteorology predictions in this study were robust to drive air quality simulation. Generally, the WRF model in this study showed a good performance, which were comparable to previous study (Shen et al., 2020b;Wang et al., 2021).

#### 3.1.2 CMAQ evaluation

The model performance of $O_3$, $NO_2$, $SO_2$, $PM_{2.5}$ and $PM_{10}$ mass in the YRD during the COVID-19 pandemic has been described in Table S2 of a previous study (Wang et al., 2020b, under review). During the whole simulated period, the predicted $PM_{2.5}$ and $O_3$ were slightly higher than observations, but the model performance was within the criteria for $PM_{2.5}$ (mean fractional bias (MFB)≤±60% and mean fractional error (MFE)≤75%, suggested by Boylan and Russell (2006)) and for $O_3$ (MFB≤±15% and MFE≤30%, suggested by U.S.EPA (2007)). Figure 1 shows predicted and observed daily $PM_{2.5}$ averaged over the YRD

and at three major cities based on Case 2 and Case 1. Generally, compared to Case 1, the lockdown significantly decreases the $PM_{2.5}$ concentration. The temporal trends of $PM_{2.5}$ mass before and during the lockdown were successfully captured by the model simulations. The MFB and MFE values of $PM_{2.5}$ mass were 0.14-0.41 and 0.38-0.57, which were all within the criteria. In Shanghai, the simulations missed the $PM_{2.5}$ episodes from January 11 to 13, but the overall performance was good. Although the overprediction was occurred both in Case 1 and Case 2, the slope of Case 2 was closer to the 1:1 line with a higher correction coefficient compared to Case 1 (Fig. S3). It indicated that the model performance was better after adjusting the emission. This discrepancy could be caused by the uncertainties in the emissions (Ying et al., 2014). The model simulation of the WRF was the same in two cases. The 2016 MEIC emission was used for the year 2020, which might overestimate the anthropogenic emissions and thus the $PM_{2.5}$ concentration in the before lockdown period. However, the emission adjustments based on Huang et al. (2020) during lockdown may be closer to the real condition, leading to better model performance. In addition, observed SIA (including sulfate, nitrate and ammonium) from January 08 to February 10 2020 in Shanghai reported by Chen et al. (2020) were used to evaluate the model performance, as shown in Fig. S4. The daily simulated trends of SIA generally agreed with the observations, although the model slightly overpredicted SIA concentrations with the MFB values of 0.19-0.37 and the MFE values of 0.41-0.68 (Table S3). The overestimation of nitrate has been reported in the previous studies (Chang et al., 2018;Shen et al., 2020b;Choi et al., 2019) and the possible reason was the lack of chlorine heterogeneous chemistry in the model (Qiu et al., 2019). Despite these uncertainties, the model results were acceptable for source apportionment studies.

### 3.2 Changes of $PM_{2.5}$ and components during the lockdown

Figure 2 shows the predicted total $PM_{2.5}$ and its components in the YRD during the COVID-19 lockdown. In both cases, $PM_{2.5}$ and its components showed similar spatial distributions with the highest concentrations in the northwest and lower concentrations in the southeast. Substantial $PM_{2.5}$ was observed in north Anhui, similar patterns were found in elemental carbon (EC) and primary organic aerosol (POA), indicating similar sources and large contributions. For Case 2, averaged $PM_{2.5}$ concentrations mainly decreased in north and west YRD due to the lockdown and all major components decreased in varying degrees. For EC and POA, similar decreases of 15% were observed in Anhui compared to Case 1. More significant decreases were found in other regions especially in Zhejiang (up to 25%). SIA had the maximum decrease in Anhui (30-40%), which was related to sharp drops of concentrations in nitrate and ammonium with decreases of 40-50% and 30-40% (Fig. S5), respectively. On the contrary, the reductions of sulfate in Shanghai were higher than other regions in the YRD, mainly due to a greater reduction of $SO_2$ from industries during the lockdown based on Huang et al. (2020). Except for central and northwest YRD, SOA decreased significantly (35-40%) also due to the reductions of industrial activities, which was an important contributor to SOA (Liu et al., 2020).

Figure 3 shows the contributions of components to $PM_{2.5}$ in the YRD and three major cities during the lockdown. For Case 2, over the entire YRD, the reductions in POA, EC, sulfate, nitrate, ammonium, and SOA was 2.4, 0.8, 2.1, 7.8, 2.9, and 0.9 µg $m^{-3}$ with a total of 17.0 µg $m^{-3}$ decrease in $PM_{2.5}$. The most significant percent decrease was found in nitrate with the highest

decrease rate of over 40%. In selected cities, PM$_{2.5}$ was decreased by 15.1, 14.8 and 16.8 μg m$^{-3}$ in Shanghai, Hangzhou and Nanjing, respectively, with the largest percent decrease of 27% in Hangzhou. Secondary components (SIA + SOA) dropped more significantly than primary components, especially for nitrate (35-45%) due to the severe decrease of NO$_x$ from transportation. This also indicated that atmospheric reactions were important during the pandemic period. In addition to nitrate, a sharp decrease was observed in ammonium due to the decrease of both nitrate and sulfate (Erisman and Schaap, 2004). SIA concentrations contributed the most to PM$_{2.5}$ in selected cities with the highest values of 26.5 μg m$^{-3}$ in Nanjing. Furthermore, the largest contributor to SIA was nitrate in the YRD, Hangzhou, and Nanjing during the lockdown, while sulfate became the dominant contributor in Shanghai and accounted for 22% of total PM$_{2.5}$, similar to the result in Chen et al. (2020).

With the impact of the lockdown, the PM$_{2.5}$ concentrations decreased significantly in the YRD region, mainly due to the reduction in the concentration of PPM and SIA. The results provided a solid basis for conducting the source apportionment of the PM$_{2.5}$ components. And the next section showed the source apportionment and regional transport of PM$_{2.5}$.

**3.3 Source sector contributions to PM$_{2.5}$**

Figure 4 shows the contributions of different source sectors to PM$_{2.5}$ in the YRD during the lockdown. Source apportionments of SIA and PPM in two cases are illustrated in Fig. S6 and Fig. S8, respectively. The agricultural source of PPM is not shown due to minor contributions. Generally, residential activities were the most significant contributor to PM$_{2.5}$ with the highest value of 45.0 μg m$^{-3}$, mainly due to the large contribution to PPM (Fig. S8). The contribution in Shanghai was ~20.0 μg m$^{-3}$ and decreased to 15.0 μg m$^{-3}$ during the lockdown. The overall decrease was less than 10% in the middle YRD and less than 15% in the rest regions. Contributions from transportation decreased the most due to the lockdown from larger than 10.0 μg m$^{-3}$ in Case 1 to less than 7.5 μg m$^{-3}$ in most areas. This is shown in SIA as well (Fig. S6), over 40% decreases were found in the YRD except for southeast, with the maximum decrease value of ~7.0 μg m$^{-3}$. The industry contributed the most to PM$_{2.5}$ values in industrial cities such as Suzhou and Hefei, positions as shown in Fig. S1, which decreased significantly by ~10.0 μg m$^{-3}$ from >30.0 μg m$^{-3}$ to ~20.0 μg m$^{-3}$ in Case 2. PM$_{2.5}$ from the power sector was decreased by less than 5% to less than 6 μg m$^{-3}$ in most areas due to reduced emissions of SO$_2$ and associated sulfate (Fig. S7). PM$_{2.5}$ from agriculture also decreased by the lockdown with the largest decrease of 5.0 μg m$^{-3}$ in northwest YRD.

Figure 5 shows the changes in contributions of sources to PM$_{2.5}$ in the YRD, Shanghai, Hangzhou, and Nanjing caused by the lockdown. Overall in the YRD, residential and industrial sources were major sources with contributions of 35% and 33% with decreases of less than 20%. Transportation, power and agriculture sources contributed similar to PM$_{2.5}$ but with different changing ratios of 40%, 6%, and 17%, respectively. Although all sources decreased, the relative contribution did not remain unchanged. The contribution ratio of transportation decreased by 27% due to the decrease in both primary emission and secondary formation, as shown in Fig. S9 and Fig. S10. The contribution ratios of residential and power increased by more than 10% while industry and agriculture showed slight changes. In large cities, industrial sources were leading with 5.0-10.0 μg m$^{-3}$ higher contribution than residential sources, while other sources were similar to the YRD averages. In Shanghai, the contributions of power and agriculture showed insignificant changes, that of the industry changed by ~20% and transportation

decreased by more than 30%. The relative contribution of transportation decreased by more than 15%, while that of power and agriculture increased by 14% and 9%, respectively. In Hangzhou and Nanjing, the trends were similar except contributions and changes of all sources were larger in Nanjing. Due to the lockdown measures, contributions of different sources decreased but their relative contribution changed differently, implying that adjustment of control measures for various sources is needed.

## 3.4 Regional contributions to $PM_{2.5}$

Figure 6 illustrates the distribution of $PM_{2.5}$ contributed by emissions from different regions for two cases in the YRD during the lockdown. Regional transmissions of SIA and PPM are shown in Fig. S11 and Fig. S12, respectively. It was clear that the regional distributions of each source were the same in both cases but Case 2 had lower values and narrower distributions. Contributions of local emissions from Jiangsu, Shanghai, Zhejiang, and Anhui generally peaked near the source regions with less than 5.0 μg m$^{-3}$ transported to other areas. Emissions from HnHb were barely transported to the central YRD area. Shandong and Ha-BTH emissions could be transported further due to north winds, as shown in Fig. S2, with ~10.0 μg m$^{-3}$ and ~5.0 μg m$^{-3}$ contributions to north YRD, respectively. It indicated that the regional transports among provinces were notable, which was consistent with Du et al. (2017). Consequently, the government should strengthen regional joint preventions in addition to local emission reductions. Other regions also had small contributions to YRD, but the contributions decreased significantly during the lockdown. The limitation of commercial activities and traffic caused by the pandemic lockdown significantly decreased the emission of $PM_{2.5}$ and indirectly suppressed its dispersion. Compared to Case 1, contributions from Jiangsu, Anhui, Shandong, and Ha-BTH in Case 2 decreased by 20-30%. More significant decreases of 30-40% were found in Shanghai, Zhejiang, and HnHb. The largest decrease of ~18.0 μg m$^{-3}$ was observed in Hubei, the center of the COVID-19 pandemic in China due to more strict lockdown measures. Figure S9 shows that after the implementation of quarantine measures, the SIA contributions decreased by more than 30% among each region and HnHb decreased by 51% to less than 10.0 μg m$^{-3}$. Figure S10 shows the narrower distributions and smaller decreases of PPM in Case 2 compared with SIA, with a decrease of less than 30% in all selected regions.

Figure 7 illustrates $PM_{2.5}$ contributed by eight regions averagely in the YRD and Shanghai. In the YRD, averaged contributions due to local emissions from Jiangsu, Shanghai, Zhejiang and Anhui were 6.8, 0.8, 1.5, and 6.3 μg m$^{-3}$ during the lockdown period, while the contribution of outside YRD areas from HnHb, Shandong, Ha-BTH and Other were 5.0, 9.1, 14.4, and 8.2 μg m$^{-3}$, respectively. The contributions of all regions decreased due to the COVID-19 lockdown with the averaged decreasing rate of 20-30% with the largest decrease rate of 33% in HnHb and the least decrease of 21% in Jiangsu. In addition to the absolute contributions, Figure 7(b) also shows the relative contribution of different regions. Ha-BTH had the largest contribution of ~30%, followed by Shandong and Other. Jiangsu and Anhui were the largest local contributors with ~12% each. It is clear that long-range transport played an important role in $PM_{2.5}$ pollution in the YRD with more than 70% contribution. Due to the COVID-19, although the absolute contributions decreased universally, their relative contributions did not. The importance of Jiangsu and Shandong increased by ~5%, while that of Shanghai, Zhejiang and HnHb decreased with the largest rate of 12% in HnHb. The results showed that although all regions reduced their concentrations to the YRD, the

relative contribution changed. In the future, regional cooperative control is needed for the YRD and strategies should be adjusted according to changes in contributions.

At the city level, local emission was the major contributor with contributions of 10.0 μg m$^{-3}$ within the YRD to Shanghai, the largest city in the YRD (Fig. 7(c)). Jiangsu contributed 16% to Shanghai while Zhejiang and Anhui had few effects. Outside the YRD, Shandong had the largest contribution (11.5 μg m$^{-3}$), followed by Ha-BTH and Other areas. In total, contributions from neighbor provinces (<10.0 μg m$^{-3}$) were much smaller than long-range transport from outside YRD (23.7 μg m$^{-3}$). Prevailing northerly winds were a key factor in this (Fig. S2). The lockdown decreased the contributions from all regions by 20-45%, with the largest decrease from HnHb. The contribution order of different regions was unchanged but their relative contributions changed. The relative contributions of local emissions from Shanghai decreased by ~10%, while that of Shandong and Jiangsu increased by ~10%. The relative contribution of HnHb decreased by more than 20%, although the absolute changes were small.

The quarantine measures during COVID-19 lockdown reduced emissions from transportation and industrials, and the total emissions for different areas changed differently. Although PM$_{2.5}$ concentrations decreased in the whole YRD, the contributions of source sectors and regions changed differently. It highlighted the need for regional cooperative emission reduction and adjusting control strategies when significant reductions were achieved.

## 4 Conclusion

A source-oriented CMAQ model investigated the changes in contributions of source sectors and regions to PM$_{2.5}$ during the COVID-19 lockdown in the YRD. Total PM$_{2.5}$ mass decreased by more than 20% across the YRD due to decreases of 30-40% and 10-20% in secondary and primary components, respectively. The results of the source apportionment showed that the residential and industrial sources were the major sources with contributions of 35% (18.0 μg m$^{-3}$) and 33% (17.1 μg m$^{-3}$) and decreased by less than 20% due to the lockdown. Contributions from transportation decreased by 40%, which was the most significant decrease, while the decrease in power was less than 10%. The relative contribution of sources changed due to differences in source decreases. The relative contribution of transportation decreased by more than 25%, while that of residential and power increased by more than 10%, suggesting that further abatement policy should adjust control measures for various sources. Contributions from regional transport of emission outside YRD were the dominant contributor (more than 70%) to the YRD, and contributions from all regions decreased due to the lockdown. The relative contribution of each region also changed with increases in Jiangsu and Shandong (~10%) but decreases in all other regions. This implied that strengthening the regional joint preventions and control of transported pollution from heavily polluted regions could effectively mitigate PM$_{2.5}$ pollution in the YRD.

*Author contributions.* JM conducted the modelling and led the writing. JS assisted in writing papers. PW, SZ, YW and PW collected data and provided technical supports. GW assisted with data analysis. JC and ZH designed the study, discussed the

results, and edited the paper.

*Competing interests.* The authors declare that they have no conflict of interest.

*Financial support.* This project was funded by Institute of Eco-Chongming (ECNU-IEC-202001).

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

Table 1. Simulation scenarios during the COVID-19 period in this study based on Huang et al. (2020).

| | Province | CO | NO$_x$ | SO$_2$ | VOC | PM$_{2.5}$ | BC | OC |
|---|---|---|---|---|---|---|---|---|
| Case 1 | All | | | | No changes | | | |
| Case 2 | Beijing | 22% | 45% | 26% | 45% | 18% | 46% | 8% |
| | Tianjin | 21% | 38% | 20% | 41% | 14% | 22% | 6% |
| | Hebei | 15% | 45% | 16% | 36% | 12% | 17% | 5% |
| | Shanxi | 18% | 40% | 20% | 33% | 16% | 19% | 10% |
| | Inner Mongolia | 14% | 29% | 15% | 34% | 13% | 16% | 6% |
| | Liaoning | 21% | 40% | 28% | 36% | 16% | 28% | 8% |
| | Jilin | 16% | 39% | 23% | 34% | 13% | 18% | 5% |
| | Heilongjiang | 17% | 37% | 27% | 28% | 13% | 15% | 7% |
| | Shanghai | 35% | 48% | 42% | 45% | 34% | 54% | 42% |
| | Jiangsu | 23% | 50% | 26% | 41% | 16% | 35% | 7% |
| | Zhejiang | 41% | 50% | 29% | 45% | 30% | 49% | 20% |
| | Anhui | 14% | 56% | 22% | 31% | 11% | 22% | 4% |
| | Fujian | 29% | 51% | 30% | 42% | 19% | 31% | 7% |
| | Jiangxi | 24% | 53% | 21% | 43% | 19% | 30% | 9% |
| | Shandong | 23% | 50% | 25% | 39% | 19% | 35% | 9% |
| | Henan | 23% | 57% | 22% | 41% | 18% | 35% | 8% |
| | Hubei | 19% | 55% | 23% | 35% | 16% | 23% | 10% |
| | Hunan | 22% | 51% | 25% | 36% | 20% | 24% | 15% |
| | Guangdong | 38% | 50% | 33% | 46% | 27% | 42% | 13% |
| | Guangxi | 24% | 50% | 28% | 39% | 17% | 27% | 5% |
| | Hainan | 24% | 44% | 25% | 36% | 14% | 25% | 4% |
| | Chongqing | 18% | 53% | 32% | 37% | 14% | 20% | 4% |
| | Sichuan | 16% | 50% | 27% | 33% | 9% | 15% | 3% |
| | Guizhou | 24% | 39% | 25% | 30% | 22% | 25% | 20% |
| | Yunnan | 24% | 51% | 25% | 41% | 18% | 21% | 8% |
| | Tibet | 16% | 35% | 15% | 35% | 14% | 14% | 5% |
| | Shaanxi | 19% | 45% | 18% | 34% | 13% | 22% | 5% |
| | Gansu | 13% | 47% | 16% | 29% | 9% | 13% | 3% |
| | Qinghai | 23% | 46% | 22% | 39% | 20% | 20% | 7% |
| | Ningxia | 24% | 36% | 24% | 39% | 20% | 23% | 8% |
| | Xinjiang | 16% | 35% | 15% | 35% | 14% | 14% | 5% |

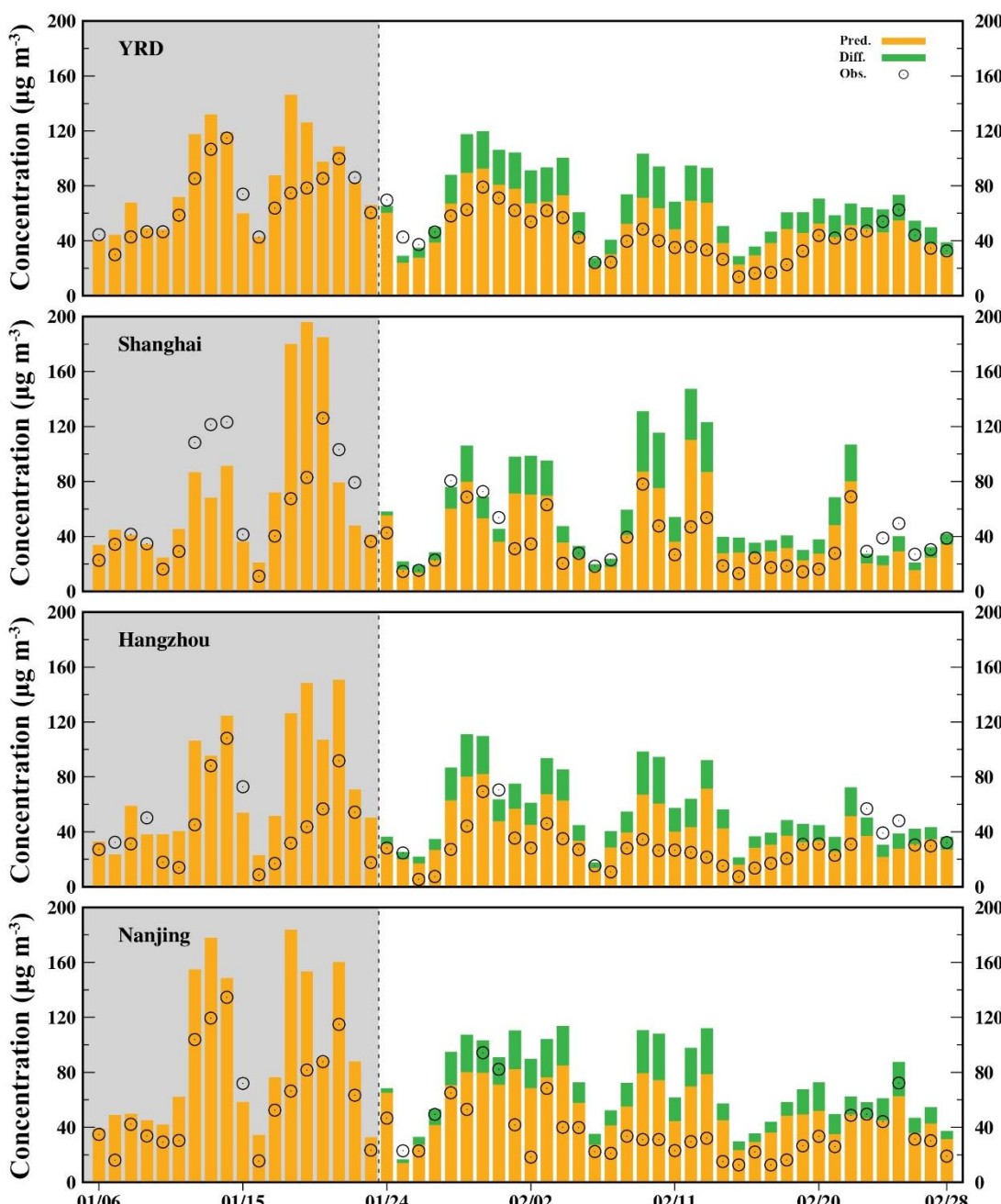

**Figure 1:** Predicted daily PM$_{2.5}$ with observed daily PM$_{2.5}$ in the YRD and three major cities in Case 2 (orange histogram) before (shaded area) and during the lockdown period (white area), the green histogram (Diff.) represents concentration difference of PM$_{2.5}$, which is calculated by Case 1 - Case 2. Units are μg m$^{-3}$. Pred. is the predicted PM$_{2.5}$ concentration, Obs. is the observed PM$_{2.5}$ concentration.

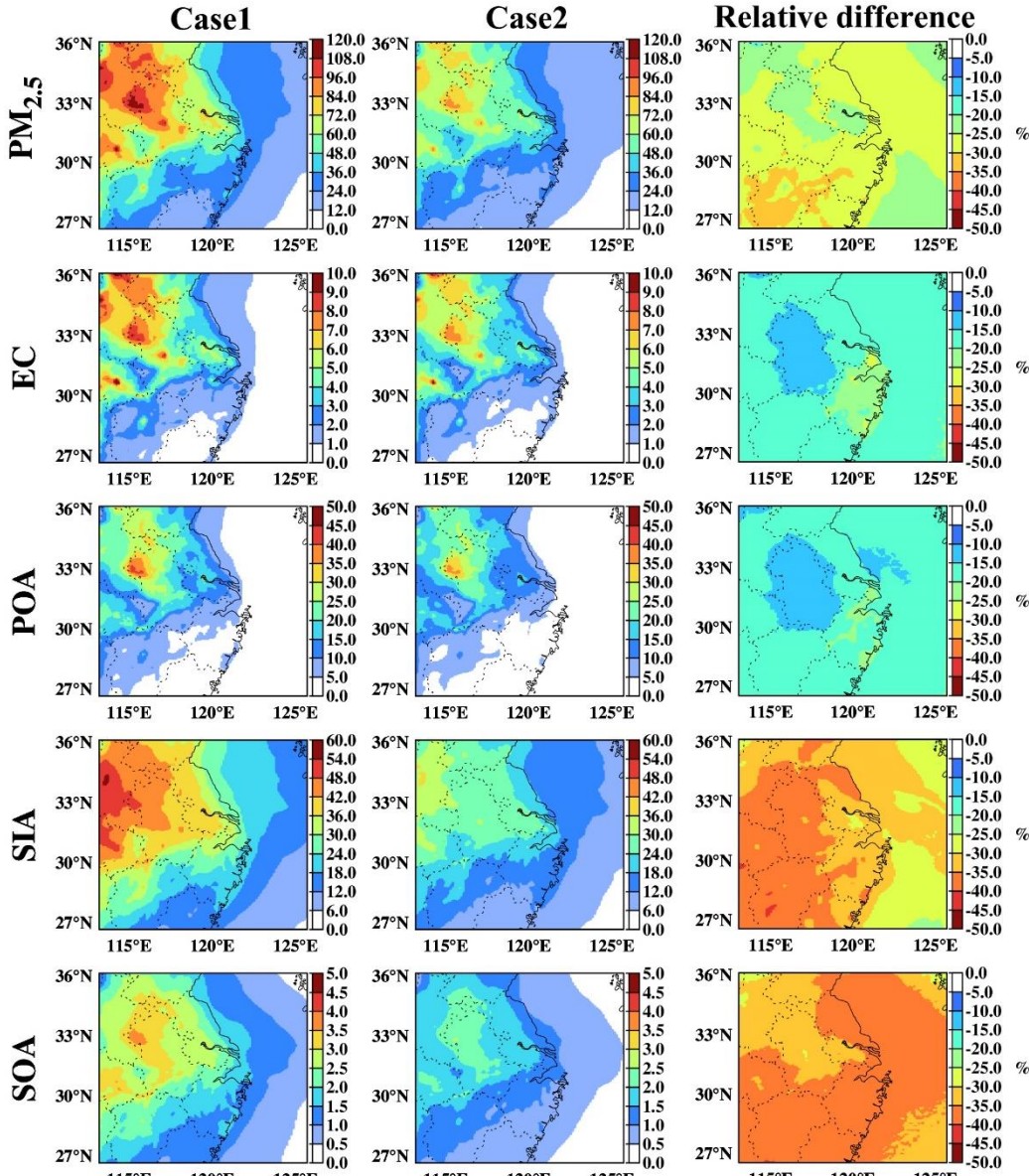

**Figure 2: Spatial distribution of predicted PM₂.₅ total and major components and changes caused by the lockdown measures in the YRD from January 23 to February 28, 2020. EC is elemental carbon, POA is primary organic aerosol. The relative difference is calculated by (Case 2 – Case 1) / Case 1, using the concentration. Note color ranges are different among panels.**

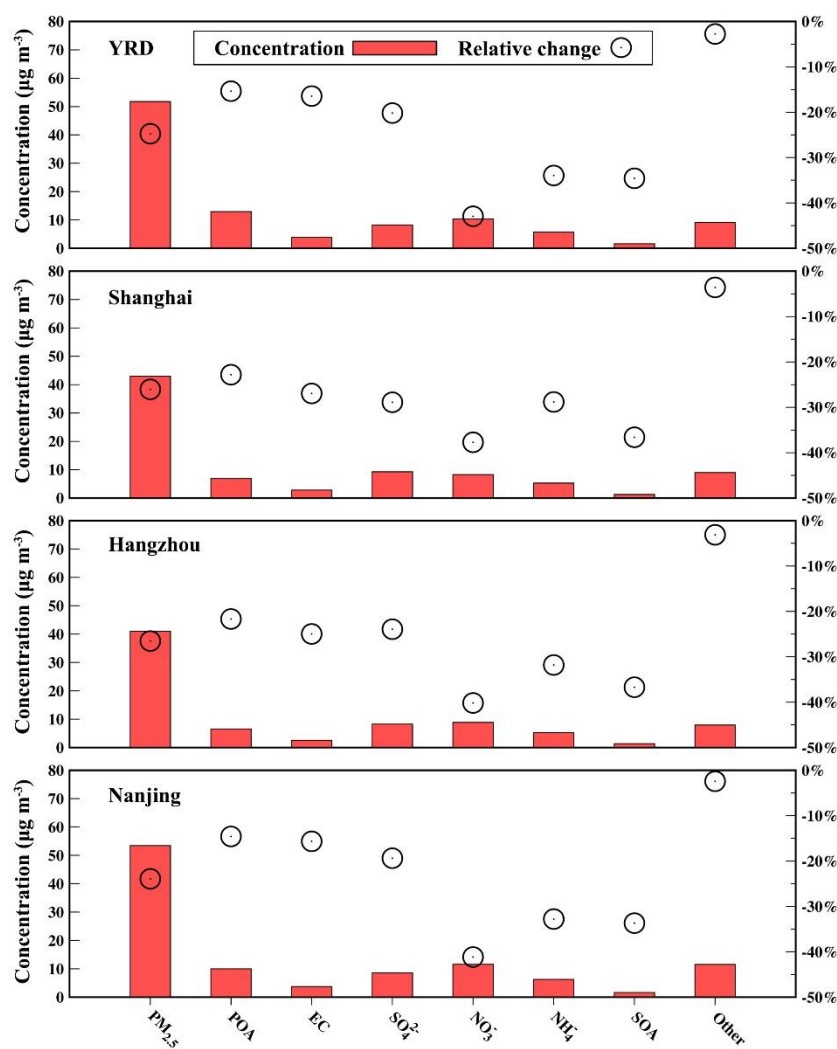


**Figure 3: Predicted PM$_{2.5}$ and its major components of Case 2 (red histogram corresponding to left Y-axis) and the relative change (circle corresponding to right Y-axis) from January 23 to February 28, 2020 in the YRD and Shanghai, Hangzhou, and Nanjing. Here the relative change means the relative change of concentration between Case 1 and Case 2, which is calculated by (Case 2 – Case 1) / Case 1.**

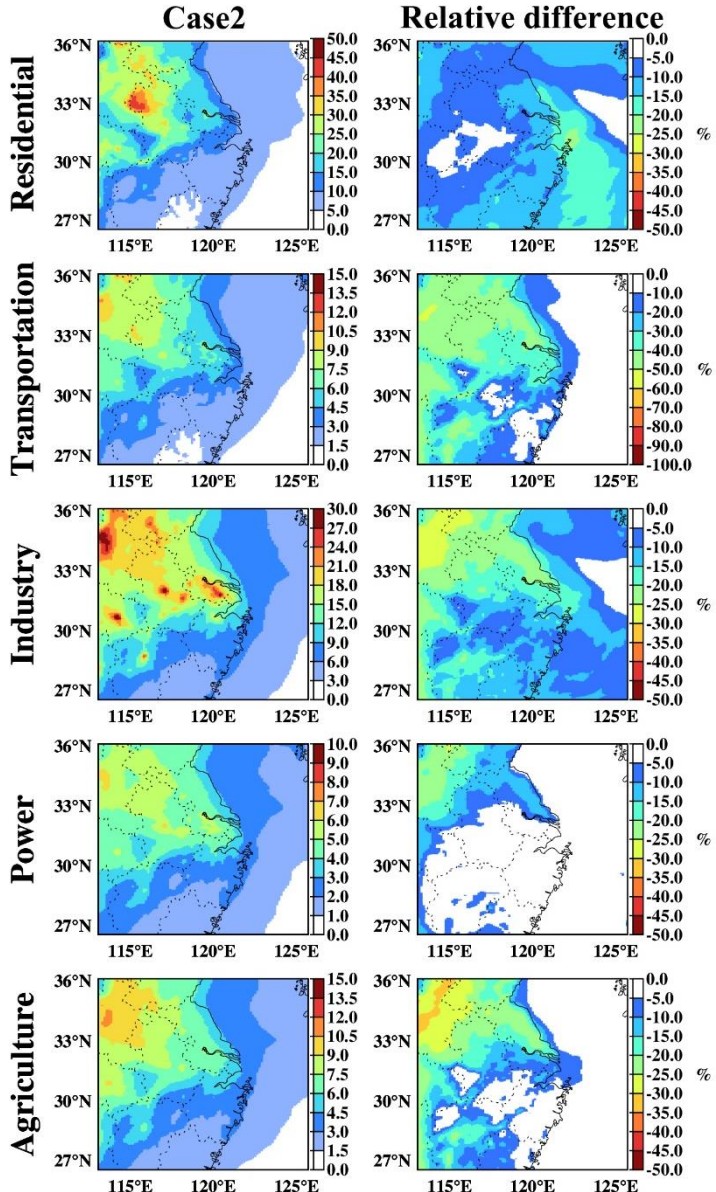


**Figure 4: Predicted PM$_{2.5}$ from different source sectors of two cases and the relative difference in the YRD from January 23 to February 28, 2020. Note color ranges are different among panels.**

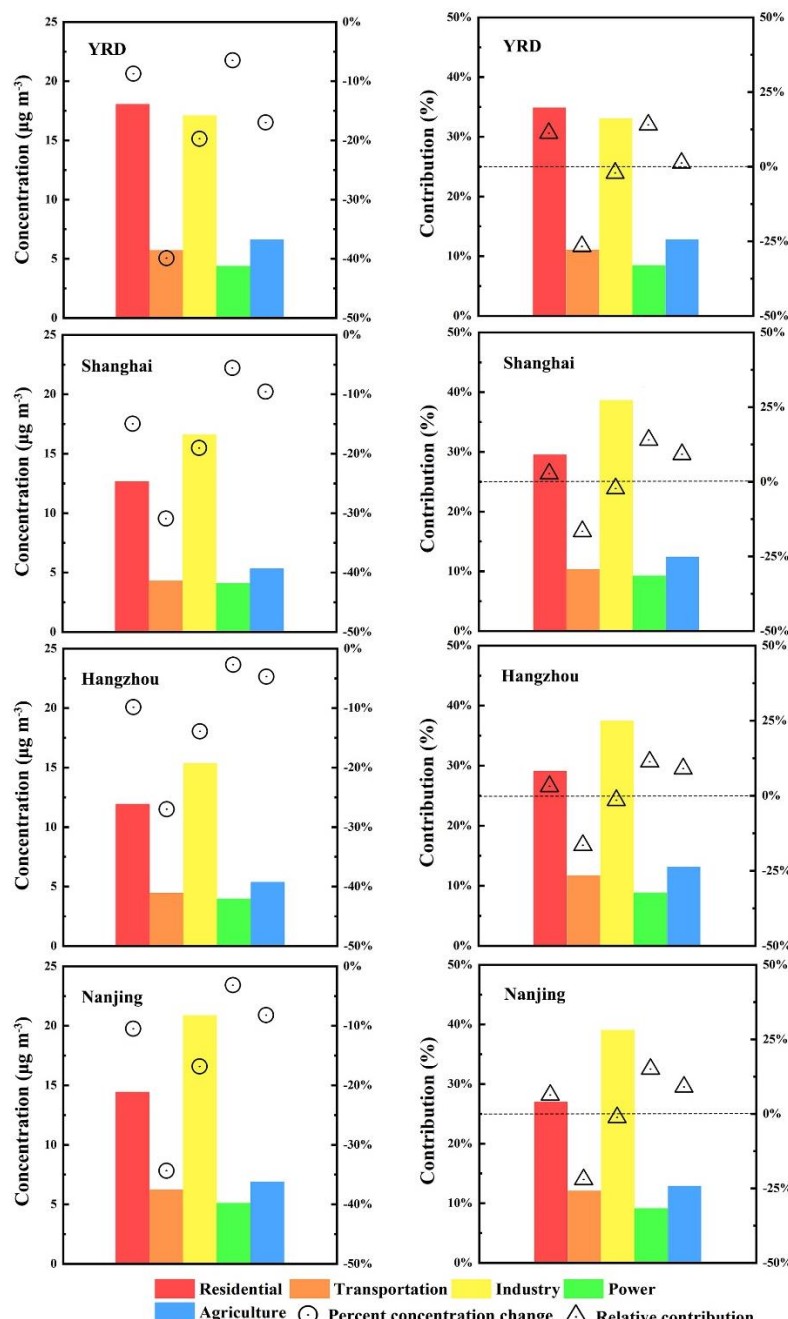

**Figure 5: Concentrations and contributions of different emission sectors to PM₂.₅ in the YRD and three major cities of Case 2 from January 23 to February 28, 2020. The values of histograms correspond to the left Y-axis and the values of relative changes correspond to the right Y-axis. The relative contribution means the relative change of contribution between Case 1 and Case 2, calculated by (Case 2 – Case 1) / Case 1. The percent concentration change means the relative change in concentration, calculated by (Case 2 – Case 1) / Case 1.**

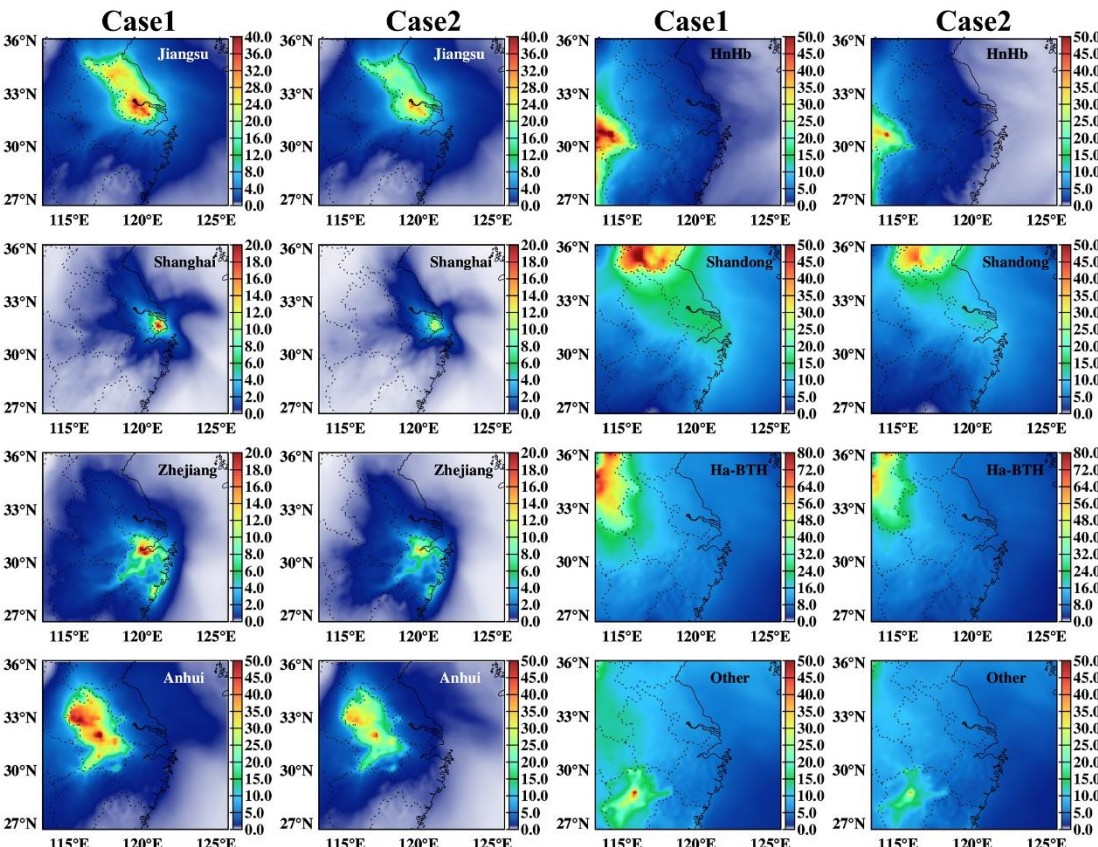

Figure 6: Averaged regional contributions of predicted PM$_{2.5}$ in the YRD from 23 to February 28, 2020. Note color ranges are different among panels.

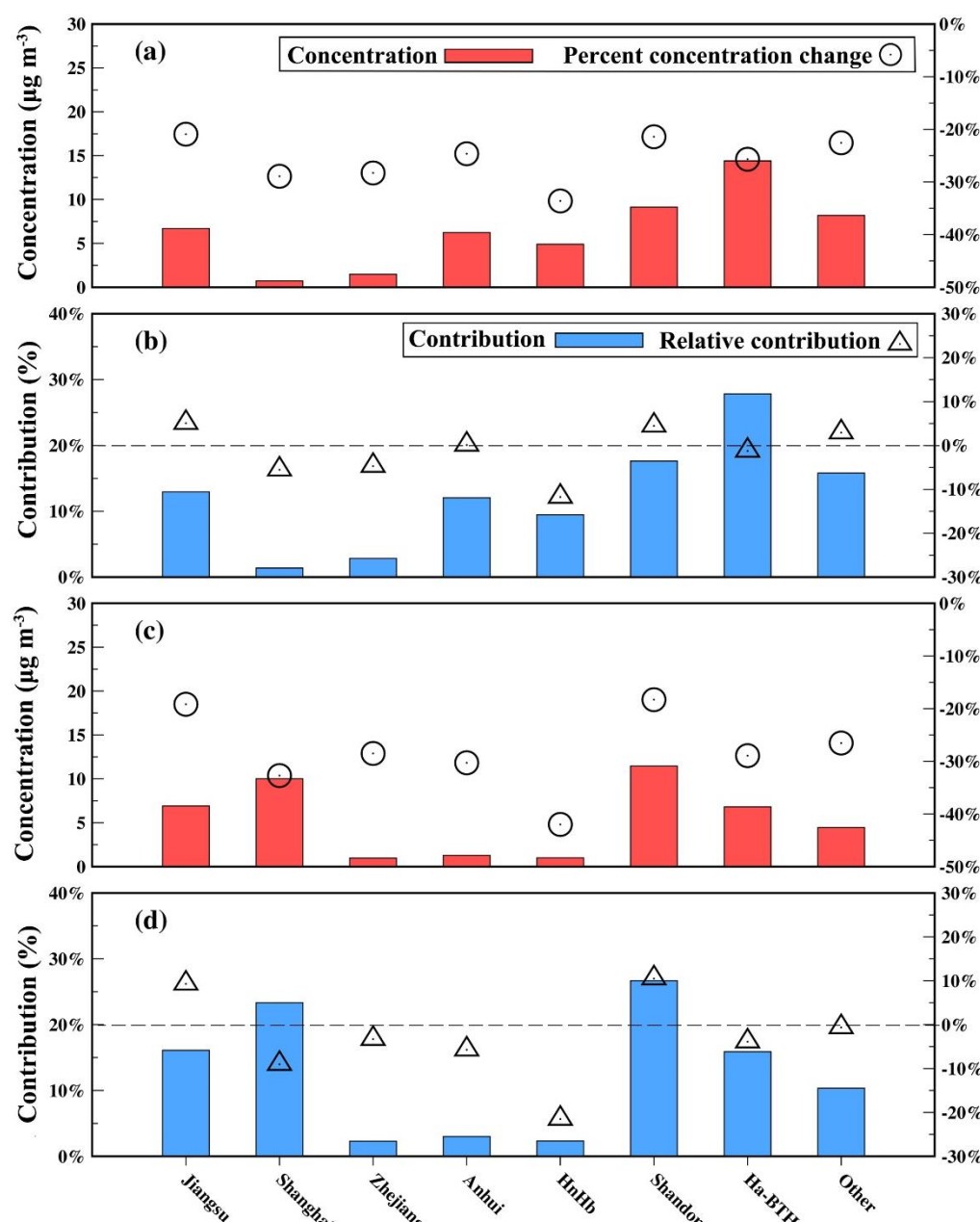

**Figure 7: Concentrations and contributions of predicted PM$_{2.5}$ from different regions in the YRD ((a) and (b)) and Shanghai ((c) and (d)) of Case 2 corresponding to left Y-axis and the relative change (corresponding to right Y-axis) from January 23 to February 28, 2020. The meanings of relative contribution and percent concentration change are the same as in Figure 5.**