# Peer review of "Modelled changes in source contributions of particulate matter during the COVID-19 pandemic in the Yangtze River Delta, China Jinlong Ma1, Juanyong Shen2, Peng Wang3, Shengqiang Zhu1, Yu Wang1, Pengfei Wang4, Gehui Wang5,6,"

_Atmospheric Chemistry and Physics, 2020_

## Referee Comment (RC1)

This paper applied the modified CMAQ model to investigate the source contribution of $PM_{2.5}$ in the nationwide lockdown due to the COVID-19 in the Yangtze River Delta region in China. By setting case 2 with reduced emission as input during the target period, the changes of source sectors of $PM_{2.5}$ were investigated, as well as the contribution of regional transport. The results showed how the contribution of location sources and regional transport changed in response to the local emission reduction and regional reduction, which are helpful to understand how regional emission reduction, quarantine measures, for example, impact the local $PM_{2.5}$ pollution and how to formula reduction policy for a more effective air quality improvement.

One of my confusions is the application of non-reactive tracer to scale the contribution of the specific source of primary PM2.5 (PPM). 0.001% was given as the ratio of the emission rate of tracers to the total PPM emission rates. Maybe I missed some key points, but I am wondering that:

1. Does the emission rate of each tracer account for 0.001% of the source emission rate? Or do the emission rates of all the tracers added up together account for 0.001% of the emission rates of all the sources?

2. Was the scaling factor $10^5$ applied for each of the sources? How the authors addressed the discrepancy between multiple sources in terms of the relative emission rate of tracers?

3. Was 0.001% a rounded assumption? If the real numbers were 0.0005% and 0.001%, both of which can be rounded to 0.001%, the scaling factors may differ by a factor of 2. I am confused so please explain.

Another concern is about the importance of revealing the change of relative contributions of regions due to the quarantine measures and its significance for future policymaking. In the discussion about the changes of regional contribution to local $PM_{2.5}$ concentration, the authors stated that the transportations of emissions were likely due to the north wind (or a key factor). If meteorology was the main reason for the change of relative contribution, what is the information the government can use when establishing the policy and adjusting the control measures? A greater reduction of emission in the regions that will contribute more to the YRD region? Please clarify the significance.

Below are several minor comments about the spelling, grammar, and others. Edits have been highlighted.

1. Line 35. "***In*** the Yangtze River Delta (YRD), one of the largest economic centers, ….". Missing *in*.

2. Line 40. Typo. "$PM_{2.5}$ is a ***complex*** mixture of …, and its source apportionment ***is*** based on quantifying the contribution of different sources to all the components.

3. Line 45. "… that local emissions ***accounted*** for the highest fraction of PPM…"

4. Line 47. "…, and ***the*** remaining contributions ***were*** from industrial and residential sector…"

5. Line 51. "…to support the **formulation** of further reduction policy."

6. Line 53. "…anthropogenic activities **since** January 2020."

7. Line 58. ", and **the conclusions reported in the mentioned literature** cannot be used to…"

8. Line 68. "was modified **with** additional non-reactive tracers…"

9. Line 72. "Details **were** discussed in Hu et al. (2015)"

10. Line 103. "The model performance of meteorological parameters **including** temperature at 2 m…"

11. Line 105. Commas should be used instead of semicolons in this sentence.

12. Line 107. "**T2 predicted by the WRF model were slightly higher** than the observations…"

13. Line 144. "More **significant** decreases were found…"

14. Line 148. "…mainly due to **a greater reduction of** $SO_2$ from industries…"

15. Line 155-157. Sentences stated that the reduction of secondary components was greater than that of primary components, which indicated the important role of atmospheric reactions and meteorological conditions. I don't see any meteorological impact was mentioned in this paragraph so please explain why the meteorological condition was important in the discussion here.

16. Line 162-163. It might better to summary the main conclusion of this section in the last few sentences and state that the analysis provides a solid basis for the later source contribution analysis. The current statement sounds like the $PM_{2.5}$ component analysis is useless for the establishment of regional emission control policies.

17. Line 193. "…were the same in both cases **but** Case 2 had lower…"

18. Line 199-200. My understanding about the regional transport of PM2.5 is mainly due to meteorology as stated in Line 196 and Line 224. So please explain why the limitation of commercial activities and traffics suppressed the dispersion of $PM_{2.5}$.

---

## Author Comment (AC1)

**Responses to interactive comments**

**Journal: Atmospheric Chemistry and Physics**

**Manuscript ID: acp-2020-953**

**Title: "Changes in source contributions of particulate matter during COVID-19 pandemic in the Yangtze River Delta, China"**

Dear Referee #1,

We appreciate your comments to help improve the manuscript. We tried our best to address your comments and detailed responses and related changes are shown below. Our response is in blue and the modifications in the manuscript are in red.

**Comments:** This paper applied the modified CMAQ model to investigate the source contribution of $PM_{2.5}$ in the nationwide lockdown due to the COVID-19 in the Yangtze River Delta region in China. By setting case 2 with reduced emission as input during the target period, the changes of source sectors of $PM_{2.5}$ were investigated, as well as the contribution of regional transport. The results showed how the contribution of location sources and regional transport changed in response to the local emission reduction and regional reduction, which are helpful to understand how regional emission reduction, quarantine measures, for example, impact the local $PM_{2.5}$ pollution and how to formula reduction policy for a more effective air quality improvement.

**Response**: Thanks for the recognition of our study. Below is the response to each specific comment.

**Comments:** One of my confusions is the application of non-reactive tracer to scale the contribution of the specific source of primary $PM_{2.5}$ (PPM). 0.001% was given as the ratio of the emission rate of tracers to the total PPM emission rates. Maybe I missed some key points, but I am wondering that:

1. Does the emission rate of each tracer account for 0.001% of the source emission rate? Or do the emission rates of all the tracers added up together account for 0.001% of the emission rates of all the sources?

**Response**: The emission rate of each tracer in each grid cell is set as 0.001% of primary $PM_{2.5}$ (PPM) emission rate from the source sector it represents. This makes sure that these tracers are small enough to not influence physical and chemical processes in the CMAQ model (Hu et al., 2015;Guo et al., 2017).

2. Was the scaling factor $10^5$ applied for each of the sources? How the authors addressed the discrepancy between multiple sources in terms of the relative emission rate of tracers?

**Response**:

Yes, the tracers were set as 0.001% of the source it represents, a factor of $10^5$ should be used to covert to the actual contribution from that source. Tracers represent different sources and have different values. Each source has its own tracer and is converted back to represent its contribution after model simulation. For instance, if the emission rate of PPM from source 1 is 1 g $s^{-1}$, the emission rate of the tracer $ATCR_1$ will be set to $1 \times 10^{-5}$ g $s^{-1}$. Similarly, if the emission rate of PPM from source 2 is 2 g $s^{-1}$, the emission rate of the tracer $ATCR_2$ will be set to $2 \times 10^{-5}$ g $s^{-1}$ (Guo et al., 2017;Hu et al., 2015). To make it clearer, the calculation formula was added into manuscript and shown below.

**Changes in manuscript:**

**Methodology (Lines 72-75 in the revision)**: "as shown in equation (1):
$$ATCR_i = 10^{-5} * PPM_i \qquad\qquad (1)$$
where $ATCR_i$ represents emission rate of the tracer from the $i$th emission source or region with PPM emission rate of $PPM_i$, and $10^{-5}$ is the scaling factor.".

3. Was 0.001% a rounded assumption? If the real numbers were 0.0005% and 0.001%, both of which can be rounded to 0.001%, the scaling factors may differ by a factor of 2. I am confused so please explain.

**Response**:

The scaling factor is an exact number, which is 0.001% for all types.

**Comments:** Another concern is about the importance of revealing the change of relative contributions of regions due to the quarantine measures and its significance for future policymaking. In the discussion about the changes of regional contribution to local $PM_{2.5}$ concentration, the authors stated that the transportations of emissions were likely due to the north wind (or a key factor). If meteorology was the main reason for the change of relative contribution, what is the information the government can use when

establishing the policy and adjusting the control measures? A greater reduction of emission in the regions that will contribute more to the YRD region? Please clarify the significance.

**Response**:

Thanks for your comments. Wind is the reason for transport of pollutants from one location to another, and north wind causes transport from Shandong and Ha-BTH to north YRD in the study period. However, this does not mean it is the main reason for the changes of relative contribution. Wind transports pollution no matter the emissions are reduced or not, but it transports different amounts of pollutants from one region (the origin) to another (the target) when the emissions from the origin region change. The different amounts of pollutants in the target region have changed relative importance. Whether it is more or less important depends on the changes of emissions from other regions.

Although this study shows the changes are important, more studies are needed to quantify the contributions from one region to another for longer period to be directly used for policy makers. This study highlights the importance of joint efforts from different regions.

We added more explanation in the manuscript to avoid misunderstanding.

**Changes in manuscript:**

**Results and discussion (Lines 206-207 in the revision)**: "Consequently, the government should strengthen regional joint preventions in addition to local emission reductions.".

**Comments:** Below are several minor comments about the spelling, grammar, and others. Edits have been highlighted.

1. Line 35. "***In*** the Yangtze River Delta (YRD), one of the largest economic centers, ….". Missing *in*.

**Response**:

The sentence was modified as suggested.

**Changes in manuscript:**

**Introduction (Line 35 in the revision):** "In the Yangtze River Delta (YRD), one of the largest economic centers, …".

2. Line 40. Typo. "$PM_{2.5}$ is a ***complex*** mixture of …, and its source apportionment ***is*** based on quantifying the contribution of different sources to all the components.

**Response**:

Mistakes were revised and shown below.

**Changes in manuscript:**

**Introduction (Line 40 in the revision):** "PM$_{2.5}$ is a complex mixture of primary PM components (PPM) and secondary formed components, …".

3. Line 45. "… that local emissions *accounted* for the highest fraction of PPM…"

**Response**:

Changed as suggested.

**Changes in manuscript:**

**Introduction (Line 47 in the revision):** "… local emissions accounted for the highest fraction of PPM …".

4. Line 47. "…, and *the* remaining contributions *were* from industrial and residential sector…"

**Response**:

We revised the sentence.

**Changes in manuscript:**

**Introduction (Line 49 in the revision):** "… and the remaining contributions were from industrial and residential sectors in Shanghai.".

5. Line 51. "…to support the *formulation* of further reduction policy."

**Response**:

Revised as below.

**Changes in manuscript:**

**Introduction (Line 53 in the revision):** "… is needed to support the formulation of further reduction policy.".

6. Line 53. "…anthropogenic activities *since* January 2020."

**Response**:

Revised accordingly.

**Changes in manuscript:**

**Introduction (Line 55 in the revision):** "… to limit anthropogenic activities since January 2020.".

7. Line 58. ", and *the conclusions reported in the mentioned literature* cannot be used to…"

**Response**:

The sentence was revised.

**Changes in manuscript:**

**Introduction (Line 60 in the revision):** "… and the conclusions reported in the mentioned literature cannot be used to design control strategies.".

8. Line 68. "was modified *with* additional non-reactive tracers…"

**Response**:

Revised as below.

**Changes in manuscript:**

**Methodology (Line 70 in the revision):** "… was modified with additional non-reactive tracers …".

9. Line 72. "Details *were* discussed in Hu et al. (2015)"

**Response**:

We revised the sentence.

**Changes in manuscript:**

**Methodology (Line 77 in the revision):** "Details were discussed in Hu et al. (2015).".

10. Line 103. "The model performance of meteorological parameters *including* temperature at 2 m…"

**Response**:

Revised accordingly.

**Changes in manuscript:**

**Results and discussion (Lines 109-110 in the revision):** "The model performance of meteorological parameters including temperature at 2 m above …".

11. Line 105. Commas should be used instead of semicolons in this sentence.

**Response**:

Changed as suggested.

**Changes in manuscript:**

**Results and discussion (Lines 111-112 in the revision):** "The statistical values of mean prediction (PRE), mean observation (OBS), mean bias (MB), gross error (GE), and root mean square error (RMSE) …".

12. Line 107. "***T2 predicted by the WRF model were slightly higher*** than the observations…"

**Response**:

Sentence was modified and shown below.

**Changes in manuscript:**

**Results and discussion (Line 113 in the revision):** "T2 predicted by the WRF model were slightly higher than observations in the two periods.".

13. Line 144. "More ***significant*** decreases were found…"

**Response**:

Revised as suggested.

**Changes in manuscript:**

**Results and discussion (Lines 151-152 in the revision):** "More significant decreases were found in …".

14. Line 148. "…mainly due to ***a greater reduction of*** SO2 from industries…"

**Response**:

Revised accordingly.

**Changes in manuscript:**

**Results and discussion (Lines 154-155 in the revision):** "… mainly due to a greater reduction of $SO_2$ from …".

15. Line 155-157. Sentences stated that the reduction of secondary components was greater than that of primary components, which indicated the important role of atmospheric reactions and meteorological

conditions. I don't see any meteorological impact was mentioned in this paragraph so please explain why the meteorological condition was important in the discussion here.

**Response**:

We are grateful for your comment. We are sorry for being not clear here. In this sentence, we intended to express that the reduction difference between secondary components and primary components was influenced by atmospheric reactions. Since, these processes were based on same meteorological conditions, there is no need to consider the impact of meteorological conditions here. We deleted the words meteorological conditions in the sentence.

**Changes in manuscript:**

**Results and discussion (Lines 162-164 in the revision):** "Secondary components (SIA + SOA) dropped more significantly than primary components, especially for nitrate (35-45%) due to the severe decrease of $NO_x$ from transportation. This also indicated that atmospheric reactions were important during the pandemic period.".

16. Line 162-163. It might better to summary the main conclusion of this section in the last few sentences and state that the analysis provides a solid basis for the later source contribution analysis. The current statement sounds like the $PM_{2.5}$ component analysis is useless for the establishment of regional emission control policies.

**Response**:

Thanks for your suggestion. The conclusion was added in the last of the section and shown below.

**Changes in manuscript:**

**Results and discussion (Lines 169-171 in the revision):** "With the impact of the lockdown, the $PM_{2.5}$ concentrations decreased significantly in the YRD region, mainly due to the reduction in the concentration of PPM and SIA. The results provided a solid basis for conducting the source appointment of the $PM_{2.5}$ components. And the next section showed the source appointment and regional transport of $PM_{2.5}$.".

17. Line 193. "…were the same in both cases *but* Case 2 had lower…"

**Response**:

Thanks for the comment. Revised as below.

**Changes in manuscript:**

**Results and discussion (Line 201 in the revision):** "… were the same in both cases but Case 2 had lower …".

18. Line 199-200. My understanding about the regional transport of $PM_{2.5}$ is mainly due to meteorology as stated in Line 196 and Line 224. So please explain why the limitation of commercial activities and traffics suppressed the dispersion of $PM_{2.5}$.

**Response**:

We appreciate the comment. As the reviewer stated, the transport of $PM_{2.5}$ was directly influenced by meteorology, but the transported concentration was also indirectly influenced by the local emission rate. Compare to Case 1 (business as usual), the transported concentration of $PM_{2.5}$ in Case 2 (lockdown) was lower under the same meteorology as shown in Figure 6 in the manuscript. This was because that the local emission rate of $PM_{2.5}$ in Case 2 was lower than that in Case 1. The lockdown measures limited the activities of commercial and traffic, which resulted in a significant decrease in $PM_{2.5}$. It also indirectly suppressed the dispersion of $PM_{2.5}$. To avoid unnecessary confusion, we added the word "indirectly" in this sentence to express more clearly and as shown below.

**Changes in manuscript:**

**Results and discussion (Lines 208-209 in the revision):** "The limitation of commercial activities and traffic caused by the pandemic lockdown significantly decreased the emission of $PM_{2.5}$ and indirectly suppressed its dispersion.".

**Reference**

Guo, H., Kota, S. H., Sahu, S. K., Hu, J., Ying, Q., Gao, A., and Zhang, H.: Source apportionment of PM2.5 in North India using source-oriented air quality models, Environ Pollut, 231, 426-436, 10.1016/j.envpol.2017.08.016, 2017.

Hu, J., Wu, L., Zheng, B., Zhang, Q., He, K., Chang, Q., Li, X., Yang, F., Ying, Q., and Zhang, H.: Source contributions and regional transport of primary particulate matter in China, Environ Pollut, 207, 31-42, 10.1016/j.envpol.2015.08.037, 2015.

---

## Author Comment (AC2)

**Responses to interactive comments**

**Journal: Atmospheric Chemistry and Physics**

**Manuscript ID: acp-2020-953**

**Title: "Changes in source contributions of particulate matter during COVID-19 pandemic in the Yangtze River Delta, China"**

Dear Referee #2,

We gratefully thank you for the constructive suggestions to help improve the manuscript. We tried our best to address your comments and detailed responses and related changes are shown below. Our responses are in blue and the modifications in the manuscript are in red.

**General Comments:**

Ma et al. present a model analysis of changes in $PM_{2.5}$ during the lockdown period associated with the COVID-19 pandemic in the Yangtze River Delta, China. The model is compared to observations of total $PM_{2.5}$ in the region, and to a more limited set of speciated $PM_{2.5}$ data from a specific site. The topic is timely and of interest to the air quality and atmospheric chemistry communities, and warrants publication in ACP.

The paper is mainly a modelling study, with relatively little reference to observations. The authors can do more to make the effects of the lockdown on $PM_{2.5}$ clear from their model based analysis. Recommendations are below in the specific comments. The word "model" should appear in the title, as it is not clear to the reader until well into the paper that all of the analysis and attributed changes are based on the model rather than any analysis of the observations. A more appropriate title would be something like "Modelled changes in source contributions of particulate matter …. "

The authors should pay attention to the specific comments and technical corrections below. They may wish to have a native English speaker proofread the paper for grammatical corrections, although the writing itself is certainly clear.

**Response**: Thanks for the recognition of our study and the good suggestion. We are sorry for the unclear expression in the title. Therefore, the title was modified in the manuscript as recommended. Below is the response to each specific comment and technical corrections.

**Changes in manuscript:**

**(Lines 1-2 in the revision): "**Modelled changes in source contributions of particulate matter during the COVID-19 pandemic in the Yangtze River Delta, China.".

**Specific Comments:**

1. Line 41-42: Why do statistical methods only address primary PM?

**Response**:

Statistical methods such as PMF (Positive Matrix Factor) and CMB (Chemical Mass Balance) use the profiles of primary emissions from different sources and assume the composition remains unchanged in the atmosphere, thus they only resolve contributions of different source sectors to PPM, leaving secondary components as a whole (Tao et al., 2014;Gao et al., 2016;Yao et al., 2016;Zhang et al., 2013). The sentence was modified to be clear.

**Changes in manuscript:**

**Introduction (Lines 41-44 in the revision)**: "Statistical methods based on observed $PM_{2.5}$ composition information using source profiles of different emission sources and assuming that composition remains unchanged in the atmosphere can only resolved contributions of different source sectors to PPM, leaving secondary components as a whole (Tao et al., 2014;Gao et al., 2016;Yao et al., 2016;Zhang et al., 2013;Zhu et al., 2018). ".

2. Line 67: The term "PM" is used with primary sources (PPM), but the term aerosol is used with secondary sources (SIA). Suggest choosing either PM or aerosol, but not mixing the two.

**Response:**

The term PPM and SIA are abbreviations of primary particulate matters and secondary inorganic aerosols, which have been widely used in previous studies (Banzhaf et al., 2013;Du et al., 2020;Guo et al., 2020;Guth et al., 2016;Hu et al., 2016;Huang et al., 2014a;Sun et al., 2016;Wang et al., 2019;Yu et al., 2018;Zhang et al., 2014). Although looked confusing, they are commonly used in numerous publications. We hope the academic field have a discussion soon to select unified usages. Therefore, we keep the abbreviation of PPM and SIA.

3. Line 79-81: The authors probably mean "due to considerable uncertainties". Beyond the grammar, however, it is difficult to believe that SOA is <10% of PM$_{2.5}$. Huang et al. 2014 (given below, also in the reference section) show that organic matter accounts for 48% of PM in Shanghai. Is there a more recent reference showing a much smaller contribution of SOA?

Huang, R.J., et al., High secondary aerosol contribution to particulate pollution during haze events in China. Nature, 2014. 514(7521): p. 218-222.

**Response**:

Thanks for understanding, we intend to express the reason for not considering SOA is due to considerable uncertainties in SOA modelling. In current models, SOA is mostly underestimated due to inadequate knowledge of its precursors, incomprehensive formation mechanisms, and limited observations (Zhao et al., 2016a;Yang et al., 2019;Heald et al., 2005;Carlton et al., 2008). Although we have the technique to resolve its sources, we have no confidence on the simulated results and leave it for future studies.

In widely used models like CMAQ and WRF-Chem, simulated SOA accounts for less than 10% of total PM$_{2.5}$. Observations in recent years show that in Shanghai and the YRD, SOA accounts for 8.6%-22.2% to total PM$_{2.5}$ (as shown in below Table R1). It indicates that more efforts are needed to improve SOA modelling in future. By the way, in Huang et al. (2014b), organic matter contributed to 48% of PM$_{2.5}$, but organic matter includes both primary organic matter and SOA. The two things can not be compared directly.

Table R1 Detailed information of references about contributions of SOA to PM$_{2.5}$

| Year | SOA contribution to PM$_{2.5}$ | Site | Reference |
|------|--------------------------------|------|-----------|
| 2020 | 16.8% at Pudong Environmental Monitoring Station | Shanghai | (Li et al., 2020) |
| 2020 | 12.6% at Pudong Supersite; 8.6% at Dianshan Lake Supersite | Shanghai | (Jia et al., 2020) |
| 2020 | 22.0% at Fengxian campus of East China University of Science and Technology (ECUST) located at the southern edge of Shanghai | Shanghai | (Sun et al., 2020) |
| 2016 | 15.7% at East China University of Science and Technology (ECUST) | Shanghai | (Zhao et al., 2016b) |
| 2016 | 22.2% at Dian Shan Lake (DSL) air quality monitoring supersite | Shanghai | (Wang et al., 2016) |

This sentence was modified to express more exact information.

**Changes in manuscript:**

**Methodology (Lines 84-86 in the revision):** "The SOA simulation has considerable uncertainties, which were caused by the inadequate knowledge of its precursors, incomprehensive formation mechanisms in the model, and limited observations (Zhao et al., 2016a;Yang et al., 2019;Heald et al., 2005;Carlton et al., 2008). Therefore, the SOA sources are not tracked in this study.".

4. Line 115: It's not clear what acceptable means here, but the quantitative measures are given above, so suggest simply omitting the last sentence of this paragraph.

**Response**:

We are sorry for the unclear expression here. The sentence was modified and shown below.

**Changes in manuscript:**

**Result and discussion (Lines 121-122 in the revision):** "Generally, the WRF model in this study showed a good performance, which were comparable to previous study (Shen et al., 2020;Wang et al., 2021).".

5. Line 121-122: The sentence is somewhat misleading in that it implies that the figure compares observations of speciated $PM_{2.5}$ to the model output. The comparison is between observed and predicted total $PM_{2.5}$ mass.

**Response**:

We appreciate your comment. We are sorry for misleading expression. This sentence was modified to express the exact information.

**Changes in manuscript:**

**Results and discussion (Lines 128-129 in the revision):** "Figure 1 shows predicted and observed daily $PM_{2.5}$ averaged over the YRD and at three major cities based on Case 2 and Case 1.".

6. Figure 1 would be far more convincing if it showed the time series of predicted total $PM_{2.5}$ mass for case 1 (business as usual) and case 2 (lockdown), as well as the difference between the two cases. It is not obvious from looking at this figure alone that the lockdowns had any influence on $PM_{2.5}$.

**Response**:

Thanks for your suggestion. Figure R1 (named as Figure 1 in the revision) was modified to show the time-series differences between Case 1 and Case 2. As shown in Figure R1, Case 1 had higher $PM_{2.5}$

emission than Case 2 during the lockdown period, indicating the lockdown policies was notable in

decreasing the PM$_{2.5}$ concentrations.

**Changes in manuscript:**

**(Lines 400-404 in the revision)**

[Figure]

**Figure R1.** Predicted daily PM$_{2.5}$ with observed daily PM$_{2.5}$ in the YRD and three major cities in Case 2 (orange histogram) before (shaded area) and during the lockdown period (white area), the green histogram (Diff.) represents concentration difference of PM$_{2.5}$, which is calculated by Case 1 - Case 2. Units are µg m$^{-3}$. Pred. is the predicted PM$_{2.5}$ concentration, Obs. is the observed PM$_{2.5}$ concentration.

7. Line 122-123: The authors should plot predicted vs. observed PM$_{2.5}$ during each period rather than

just providing the time series for the comparison. A slope of a linear fit to this scatter plot would provide

a quantitative measure of model performance. Similarly, a slope of the case 1 prediction against the

observations would show how well this case performed prior to the lockdowns, as well as how much is overpredicted the observations during the lockdown.

**Response**:

Thanks for your suggestion. The linear fit of $PM_{2.5}$ predictions vs. observations in each period of Case 1 and Case 2 was drawn in Figure R2 (add as Figure S3 in the revision) in the supplementary material as shown below. The general agreement was found between the predicted and observed $PM_{2.5}$, with more than 90% of data points falling into the 1:2 and 2:1 dash lines in the YRD. Although the overprediction was occurred both in Case 1 and Case 2, the slope of Case 2 was closer to the 1:1 line with a higher correction coefficient compared to Case 1.

**Changes in manuscript:**

**(Lines 39-41 in the revised supplementary material)**

[Figure]

**Figure R2.** Comparisons of observed and predicted daily concentration of PM$_{2.5}$ in the YRD and three major cities of Case 1 and Case 2 in each period. R is the correction coefficient. The dash lines in the plot are 1:2, 1:1, 2:1, respectively. Unit is μg m$^{-3}$.

**Changes in manuscript:**

**Results and discussion (Lines 131-136 in the revision):** "Although the overprediction was occurred both in Case 1 and Case 2, the slope of Case 2 was closer to the 1:1 line with a higher correction coefficient compared to Case 1 (Fig. S3). It indicated that the model performance was better after adjusting the emission. This discrepancy could be caused by the uncertainties in the emissions (Ying et al., 2014). The model simulation of the WRF was the same in two cases.".

8. Figure 3: The labeling is not quite clear. It appears the authors mean "percent concentration change" rather than "relative concentration" for the circles that are plotted against the right axis.

**Response**:

Thank you for your valuable suggestion. The label "relative concentration" in Figure R3 (named as Figure 3 in the revision) was modified to "relative change", which was calculated by (Case 2- Case 1)/ Case1, and was mentioned in the caption. The modifications were also taken in Figure 5, Figure 7, Figure S9 and Figure S10 in the revision.

[Figure]

**Figure R3.** Predicted PM$_{2.5}$ and its major components of Case 2 (red histogram corresponding to left Y-axis) and the relative change (circle corresponding to right Y-axis) from January 23 to February 28, 2020 in the YRD and Shanghai, Hangzhou, and Nanjing. Here the relative change means the relative change of concentration between Case 1 and Case 2, which is calculated by (Case 2 – Case 1) / Case 1.

**Changes in manuscript:**

 "Figure 3: Predicted $PM_{2.5}$ and its major components of Case 2 (red histogram corresponding to left Y-axis) and the relative change (circle corresponding to right Y-axis) from January 23 to February 28, 2020 in the YRD and Shanghai, Hangzhou, and Nanjing. Here the relative change means the relative change of concentration between Case 1 and Case 2, which is calculated by (Case 2 – Case 1) / Case 1.".

9. Figure 4: Why is Case 1 (base case, no reductions) not also shown? It would seem the business as usual case is as important to show as the reduced emissions case. Also, all of the relative differences are negative. Why? Shouldn't the residential sector increase while transportation and industry decrease?

Perhaps what would make the above clearer is the apportionment among sources for case 1 and case 2 – i.e., what fraction of $PM_{2.5}$ is attributable to each source in each case. This measure would likely show that residential was a larger overall contributor for case 2.

**Response**:

Thanks for your comment. For the first question, we think Case 2 with the relative difference between Case 1 and Case 2 in Figure 4 of in the manuscript can also express the information of Case 1, so Case 1 is not shown. For the second question, the adjustment of emission was based on Huang et al. (2020), also as shown in Table 1. Huang et al. (2020) explained that the residential sector includes the commercial use of boilers and stoves and residential heating and cooking. During the lockdown period, the commercial use of boilers and stoves was closed, but the uses of residential heating and cooking remained the same. Therefore, the residential sector was also affected by the lockdown measures. In addition, Figure 5 in the manuscript shows the contribution from the residential sector were the major contributor to the YRD in Case 2.

10. Figure 5 is also difficult to read. The authors should consider using a pie chart format in which the contribution from each sector is shown as a wedge in a pie for case 1 and case 2. This would make clear how the sources changed between business as usual and lockdown policies.

**Response**:

We have considered using the pie chart to express the source contribution information of Case 1 and Case 2 as shown in Figure R4, but it was more difficult to visualize the change in source contribution. Thus, we keep Figure 5 in the manuscript.

[Figure]

**Figure R4. The source contribution of PM$_{2.5}$ in the YRD and three major cities in Case 1 and Case 2.**

**Technical Corrections:**

1. Title: Should read "the COVID-19 pandemic".

**Response**:

Gratefully thanks for your comment. Revised as below.

**Changes in manuscript:**

**Title (Lines 1-2 in the revision):** "Modelled changes in source contributions of particulate matter during the COVID-19 pandemic in the Yangtze River Delta, China".

2. Line 35: "In the Yangtze River Delta …"

**Response:**

Thanks for your comment. Revised accordingly.

**Changes in manuscript:**

**Introduction (Line 35 in the revision):** "In the Yangtze River Delta (YRD) …".

3. Line 41: "… is based …"

**Response:**

We show a grateful appreciation for your comment. Revised as below.

**Changes in manuscript:**

**Introduction (Lines 40-41 in the revision):** "… its source apportionment is based on quantifying …".

4. Line 50-51: Therefore, updated source apportionment information is needed to support further reduction policy.

**Response:**

We show a grateful appreciation for your comment. Revised as below.

**Changes in manuscript:**

**Introduction (Lines 52-53 in the revision):** "Therefore, updated source appointment information is needed to support the formulation of further reduction policy.".

5. Line 57-58: "changes, and these studies cannot be used …".

**Response:**

Thanks for your comment. This sentence was modified and shown below.

**Changes in manuscript:**

**Introduction (Lines 59-60 in the revision):** "and the conclusions reported in the mentioned literature cannot be used to design control strategies.".

6. Line 110: replace "were met" with "met".

**Response**:

We are grateful for your comment. Revised accordingly.

**Changes in manuscript:**

**Results and discussion (Line 116 in the revision):** "… RMSE (1.7 and 2.0) met the benchmarks …".

7. Line 121: "Figure 1 shows predicted …"

**Response**:

Thanks for your comment. Revised as suggested.

**Changes in manuscript:**

**Results and discussion (Line 128 in the revision):** "Figure 1 shows predicted …".

8. Line 146: Replace "decreasing ratios" with "decreases"

**Response**:

Thanks for the comment. Revised as below.

**Changes in manuscript:**

**Results and discussion (Line 154 in the revision):** "… with decreases of 40-50% …".

9. Line 148: Replace "reduced more" with "more reduced".

**Response**:

We appreciate your rigorous comment. This sentence was modified.

**Changes in manuscript:**

**Results and discussion (Lines 155-156 in the revision):** "… mainly due to a greater reduction of $SO_2$ from …".

10. Line 153, 155: Replace "decrease" and "decrease ratio" with "percent decrease"

**Response**:

We appreciate your rigorous comment. We revised the sentences.

**Changes in manuscript:**

**Results and discussion (Line 159 in the revision):** "The most significant percent decrease was found … the largest percent decrease of 27% …".

11. Line 163: Replace "Below" with "The next section"

**Response**:

We appreciate your rigorous comment. Revised as below.

**Changes in manuscript:**

**Results and discussion (Line 172 in the revision):** "And the next section showed the source appointment and regional transport of $PM_{2.5}$.".

12. Line 199: Replace "traffics" with "traffic"

**Response**:

We appreciate your rigorous comment. Revised accordingly.

**Changes in manuscript:**

**Results and discussion (Line 209 in the revision):** "… commercial activities and traffic …".

**Reference**

Banzhaf, S., Schaap, M., Kruit, R. J. W., van der Gon, H. A. C. D., Stern, R., and Builtjes, P. J. H.: Impact of emission changes on secondary inorganic aerosol episodes across Germany, Atmospheric Chemistry and Physics, 13, 11675-11693, 10.5194/acp-13-11675-2013, 2013.

Carlton, A. G., Turpin, B. J., Altieri, K. E., Seitzinger, S. P., Mathur, R., Roselle, S. J., and Weber, R. J.: CMAQ Model Performance Enhanced When In-Cloud Secondary Organic Aerosol is Included: Comparisons of Organic Carbon Predictions with Measurements, Environmental Science & Technology, 42, 8798-8802, 10.1021/es801192n, 2008.

Du, H., Li, J., Wang, Z., Dao, X., Guo, S., Wang, L., Ma, S., Wu, J., Yang, W., Chen, X., and Sun, Y.: Effects of Regional Transport on Haze in the North China Plain: Transport of Precursors or Secondary Inorganic Aerosols, Geophysical Research Letters, 47, 10.1029/2020gl087461, 2020.

Gao, J., Peng, X., Chen, G., Xu, J., Shi, G.-L., Zhang, Y.-C., and Feng, Y.-C.: Insights into the chemical characterization and sources of PM2.5 in Beijing at a 1-h time resolution, Science of the Total Environment, 542, 162-171, 10.1016/j.scitotenv.2015.10.082, 2016.

Guo, J., Zhou, S., Cai, M., Zhao, J., Song, W., Zhao, W., Hu, W., Sun, Y., He, Y., Yang, C., Xu, X., Zhang, Z., Cheng, P., Fan, Q., Hang, J., Fan, S., Wang, X., and Wang, X.: Characterization of submicron particles by time-of-flight aerosol chemical speciation monitor (ToF-ACSM) during wintertime: aerosol composition, sources, and chemical processes in Guangzhou, China, Atmospheric Chemistry and Physics, 20, 7595-7615, 10.5194/acp-20-7595-2020, 2020.

Guth, J., Josse, B., Marecal, V., Joly, M., and Hamer, P.: First implementation of secondary inorganic aerosols in the MOCAGE version R2.15.0 chemistry transport model, Geoscientific Model Development, 9, 137-160, 10.5194/gmd-9-137-2016, 2016.

Heald, C. L., Jacob, D. J., Park, R. J., Russell, L. M., Huebert, B. J., Seinfeld, J. H., Liao, H., and Weber, R. J.: A large organic aerosol source in the free troposphere missing from current models, Geophysical Research Letters, 32, n/a-n/a, 10.1029/2005gl023831, 2005.

Hu, W., Hu, M., Hu, W., Jimenez, J. L., Yuan, B., Chen, W., Wang, M., Wu, Y., Chen, C., Wang, Z., Peng, J., Zeng, L., and Shao, M.: Chemical composition, sources, and aging process of submicron aerosols in Beijing: Contrast between summer and winter, Journal of Geophysical Research-Atmospheres, 121, 1955-1977, 10.1002/2015jd024020, 2016.

Huang, R.-J., Zhang, Y., Bozzetti, C., Ho, K.-F., Cao, J.-J., Han, Y., Daellenbach, K. R., Slowik, J. G., Platt, S. M., Canonaco, F., Zotter, P., Wolf, R., Pieber, S. M., Bruns, E. A., Crippa, M., Ciarelli, G., Piazzalunga, A., Schwikowski, M., Abbaszade, G., Schnelle-Kreis, J., Zimmermann, R., An, Z., Szidat, S., Baltensperger, U., El Haddad, I., and Prevot, A. S. H.: High secondary aerosol contribution to particulate pollution during haze events in China, Nature, 514, 218-222, 10.1038/nature13774, 2014a.

Huang, R. J., Zhang, Y., Bozzetti, C., Ho, K. F., Cao, J. J., Han, Y., Daellenbach, K. R., Slowik, J. G., Platt, S. M., Canonaco, F., Zotter, P., Wolf, R., Pieber, S. M., Bruns, E. A., Crippa, M., Ciarelli, G., Piazzalunga, A., Schwikowski, M., Abbaszade, G., Schnelle-Kreis, J., Zimmermann, R., An, Z., Szidat, S., Baltensperger, U., El Haddad, I., and Prevot, A. S.: High secondary aerosol contribution to particulate pollution during haze events in China, Nature, 514, 218-222, 10.1038/nature13774, 2014b.

Jia, H., Huo, J., Fu, Q., Duan, Y., Lin, Y., Jin, X., Hu, X., and Cheng, J.: Insights into chemical composition, abatement mechanisms and regional transport of atmospheric pollutants in the Yangtze River Delta region, China during the COVID-19 outbreak control period, Environ Pollut, 267, 115612, 10.1016/j.envpol.2020.115612, 2020.

Li, R., Wang, Q., He, X., Zhu, S., Zhang, K., Duan, Y., Fu, Q., Qiao, L., Wang, Y., Huang, L., Li, L., and Yu, J. Z.: Source apportionment of $PM_{2.5}$ in Shanghai based on hourly organic molecular markers and other source tracers, Atmospheric Chemistry and Physics, 20, 12047-12061, 10.5194/acp-20-12047-2020, 2020.

Shen, J., Zhao, Q., Cheng, Z., Wang, P., Ying, Q., Liu, J., Duan, Y., and Fu, Q.: Insights into source origins and formation mechanisms of nitrate during winter haze episodes in the Yangtze River Delta, Sci Total Environ, 741, 140187, 10.1016/j.scitotenv.2020.140187, 2020.

Sun, P., Nie, W., Wang, T., Chi, X., Huang, X., Xu, Z., Zhu, C., Wang, L., Qi, X., Zhang, Q., and Ding, A.: Impact of air transport and secondary formation on haze pollution in the Yangtze River Delta: In situ online observations in Shanghai and Nanjing, Atmospheric Environment, 225, 10.1016/j.atmosenv.2020.117350, 2020.

Sun, Y., Wang, Z., Wild, O., Xu, W., Chen, C., Fu, P., Du, W., Zhou, L., Zhang, Q., Han, T., Wang, Q., Pan, X., Zheng, H., Li, J., Guo, X., Liu, J., and Worsnop, D. R.: "APEC Blue": Secondary Aerosol Reductions from Emission Controls in Beijing, Scientific Reports, 6, 10.1038/srep20668, 2016.

Tao, J., Gao, J., Zhang, L., Zhang, R., Che, H., Zhang, Z., Lin, Z., Jing, J., Cao, J., and Hsu, S. C.: $PM_{2.5}$ pollution in a megacity of southwest China: source apportionment and implication, Atmos. Chem. Phys., 14, 8679-8699, 10.5194/acp-14-8679-2014, 2014.

Wang, D., Zhou, B., Fu, Q., Zhao, Q., Zhang, Q., Chen, J., Yang, X., Duan, Y., and Li, J.: Intense secondary aerosol formation due to strong atmospheric photochemical reactions in summer: observations at a rural site in eastern Yangtze River Delta of China, Sci Total Environ, 571, 1454-1466, 10.1016/j.scitotenv.2016.06.212, 2016.

Wang, H., Ding, J., Xu, J., Wen, J., Han, J., Wang, K., Shi, G., Feng, Y., Ivey, C. E., Wang, Y., Nenes, A., Zhao, Q., and Russell, A. G.: Aerosols in an arid environment: The role of aerosol water content,

particulate acidity, precursors, and relative humidity on secondary inorganic aerosols, Science of the Total Environment, 646, 564-572, 10.1016/j.scitotenv.2018.07.321, 2019.

Wang, X., Li, L., Gong, K., Mao, J., Hu, J., Li, J., Liu, Z., Liao, H., Qiu, W., Yu, Y., Dong, H., Guo, S., Hu, M., Zeng, L., and Zhang, Y.: Modelling air quality during the EXPLORE-YRD campaign – Part I. Model performance evaluation and impacts of meteorological inputs and grid resolutions, Atmospheric Environment, 246, 10.1016/j.atmosenv.2020.118131, 2021.

Yang, W., Li, J., Wang, W., Li, J., Ge, M., Sun, Y., Chen, X., Ge, B., Tong, S., Wang, Q., and Wang, Z.: Investigating secondary organic aerosol formation pathways in China during 2014, Atmospheric Environment, 213, 133-147, 10.1016/j.atmosenv.2019.05.057, 2019.

Yao, L., Yang, L., Yuan, Q., Yan, C., Dong, C., Meng, C., Sui, X., Yang, F., Lu, Y., and Wang, W.: Sources apportionment of PM2.5 in a background site in the North China Plain, Science of the Total Environment, 541, 590-598, 10.1016/j.scitotenv.2015.09.123, 2016.

Ying, Q., Wu, L., and Zhang, H.: Local and inter-regional contributions to PM2.5 nitrate and sulfate in China, Atmospheric Environment, 94, 582-592, 10.1016/j.atmosenv.2014.05.078, 2014.

Yu, J., Jihan, S., Song, J., Lee, D., Yu, M., and Kim, J.: A Study on the Change of Condensable Particulate Matter by the SO2 Concentration among Combustion Gases, Journal of Korean Society for Atmospheric Environment, 34, 651-658, 10.5572/kosae.2018.34.5.651, 2018.

Zhang, R., Jing, J., Tao, J., Hsu, S. C., Wang, G., Cao, J., Lee, C. S. L., Zhu, L., Chen, Z., Zhao, Y., and Shen, Z.: Chemical characterization and source apportionment of PM2.5 in Beijing: seasonal perspective, Atmospheric Chemistry and Physics, 13, 7053-7074, 10.5194/acp-13-7053-2013, 2013.

Zhang, Y., Wang, W., Wu, S.-Y., Wang, K., Minoura, H., and Wang, Z.: Impacts of updated emission inventories on source apportionment of fine particle and ozone over the southeastern US, Atmospheric Environment, 88, 133-154, 10.1016/j.atmosenv.2014.01.035, 2014.

Zhao, B., Wang, S., Donahue, N. M., Jathar, S. H., Huang, X., Wu, W., Hao, J., and Robinson, A. L.: Quantifying the effect of organic aerosol aging and intermediate-volatility emissions on regional-scale aerosol pollution in China, Sci Rep, 6, 28815, 10.1038/srep28815, 2016a.

Zhao, M., Xiu, G., Qiao, T., Li, Y., and Yu, J.: Characteristics of Haze Pollution Episodes and Analysis of a Typical Winter Haze Process in Shanghai, Aerosol and Air Quality Research, 16, 1625-1637, 10.4209/aaqr.2016.01.0049, 2016b.

Zhu, Y., Huang, L., Li, J., Ying, Q., Zhang, H., Liu, X., Liao, H., Li, N., Liu, Z., Mao, Y., Fang, H., and Hu, J.: Sources of particulate matter in China: Insights from source apportionment studies published in 1987-2017, Environ Int, 115, 343-357, 10.1016/j.envint.2018.03.037, 2018.